# Improving the Performance of Mode-Based Sound Propagation Models by Using Perturbation Formulae for Eigenvalues and Eigenfunctions

Alena Zakharenko [ID], Mikhail Trofimov [ID] and Pavel Petrov *[ID]

V.I.Il'ichev Pacific Oceanological Institute, 43, Baltiyskaya st, 690041 Vladivostok, Russia;
zakharenko@poi.dvo.ru (A.Z.); trofimov@poi.dvo.ru (M.T.)
* Correspondence: petrov@poi.dvo.ru

**Abstract:** Numerous sound propagation models in underwater acoustics are based on the representation of a sound field in the form of a decomposition over normal modes. In the framework of such models, the calculation of the field in a range-dependent waveguide (as well as in the case of 3D problems) requires the computation of normal modes for every point within the area of interest (that is, for each pair of horizontal coordinates x,y). This procedure is often responsible for the lion's share of total computational cost of the field simulation. In this study, we present formulae for perturbation of eigenvalues and eigenfunctions of normal modes under the water depth variations in a shallow-water waveguide. These formulae can reduce the total number of mode computation instances required for a field calculation by a factor of 5–10. We also discuss how these formulae can be used in a combination with a wide-angle mode parabolic equation. The accuracy of such combined model is validated in a series of numerical examples.

**Keywords:** underwater acoustics; normal modes; perturbation theory; rough bottom; mode parabolic equations

## 1. Introduction

Computational technique based on the normal modes theory is widely used in underwater acoustics and its applications that cover a large area of marine sciences [1–3]. It is important that normal modes provide both quantitative and qualitative understanding of many physical effects related to sound propagation in the sea. Investigation of the dependence of modal wavenumbers $k_j$ and eigenfunctions $\phi_j(z)$ on certain environment factors and parameters is often sufficient for estimating the influence of these factors on the structure of acoustical field (in particular, on interference patterns, shadow zones, arrival times, and energy distribution [4,5]).

Indeed, sound field in a 3D oceanic waveguide formed by the sea surface $z = 0$ and sea bottom $z = h(x, y)$ (here $z$ stands for depth, while $x, y$ are the horizontal coordinates, see in [1] for detailed discussion of waveguides in underwater acoustics) can be represented in the form of the expansion

$$P(x, y, z) = \sum_{j=1}^{N_m} A_j(x, y)\phi_j(z, x, y) \tag{1}$$

over local normal modes $\phi_j(z, x, y)$ (i.e., the eigenfunctions $\phi_j(z)$ computed for the given values of $x, y$). Under the adiabatic assumption, mode amplitudes $A_j(x, y)$ satisfy the so-called horizontal refraction equation (HRE) [1,5,6]

$$\frac{\partial^2 A_j}{\partial x^2} + \frac{\partial^2 A_j}{\partial y^2} + k_j^2 A_j = 0, \tag{2}$$

and therefore the dependence of eigenvalue $k_j^2 = (k_j(x, y))^2$ on the variables $x, y$ is what actually determines the propagation of sound waves associated with each mode in the horizontal plane.

In this case, the distribution of $k_j(x, y)$ plays the role of effective refractive index for horizontal rays [5,7]. In many similar sound propagation models (e.g., in the mode parabolic equation theory [8–10]), variation of media parameters in the horizontal plane also influences acoustical field via the corresponding variability of horizontal wavenumbers in Equation (2) and eigenfunctions in Equation (1).

It is therefore desirable to have explicit formulae describing the dependence of eigenvalues $k_j$ and modal functions $\phi_j(z)$ on the horizontal variables $x, y$.

The variability of modes in $x$ and $y$ can result from the sound speed dependence on $x, y$. In this case, the standard perturbation theory from quantum mechanics [1,11] can be applied to take this dependence into account. For example, it is routinely used to the compute imaginary parts of eigenvalues and modal group velocities [1].

However, eigenvalues and mode functions can also depend on the horizontal coordinates due to variations of bottom relief. In shallow-water acoustics, this factor is of primary importance, especially for low-frequency sound that is less sensitive to volume inhomogeneities of the sound speed in the water column. This problem is obviously connected with differentiation of eigenvalues and mode functions with respect to the water depth $h$. Explicit formulae for the derivatives of eigenvalues with respect to range in the presence of water depth variations $h = h(x, y)$ were first derived by Brekhovskikh and Godin [12]. Later, Godin [13] also obtained expressions for the derivatives of mode functions from the so-called generalized orthogonality relationships [12]. Obviously, the problem of differentiation of eigenvalues and eigenfunctions with respect to the range $r$ can be reduced to the differentiation with respect to water depth $h$, provided that environmental gradient, i.e., $dh/dr$, is known. Identical formulae were derived independently by Trofimov [14] who used multi-scale approach. To our knowledge, however, neither a second-order perturbation theory nor explicit formulae for second-order derivatives of eigenvalues and mode functions in the case of water depth variations $h = h(x, y) = h_0 + \Delta h(x, y)$ (see Figure 1) have been presented in the literature. It appears that the derivation techniques from in [12–14] cannot be extended to the second-order case. We also observed that the convergence of Godin's series for the derivatives of mode functions is slow, and the respective approximation error decreases in a non-monotonic way. Clearly, this makes the formulae for the derivatives of modal functions from in [13,14] ill-suited for practical computations. Very recently, explicit formulae for the first and second derivatives of eigenvalues and modal functions in the case of water depth variations were outlined by Petrov et al. [15]. The approach used in the latter paper is different from those of Godin [13] and Trofimov [14]. For the first-order derivatives of eigenvalues, the resulting formulae are the same as derived by Trofimov and Godin [13,14] (modulo some simple transformations). At the same time, the numerical examples presented in this study indicate that our expressions for the derivatives of mode functions appear to be more practical and computationally robust.

The main goal of this study is to provide a detailed derivation of the formulae from in [15], including the generalization to the third-order perturbations of horizontal wavenumbers that were not given in the mentioned paper, and to show how this perturbation theory can be used in realistic problems of sound propagation. Note that in [15] only the very idea of the derivation of the perturbative formulae is given, while the calculations are largely omitted.

The paper is organized as follows. We start with the formulation of the Sturm–Liouville problem from which horizontal wavenumbers and mode functions are determined (Section 2). In Section 3, we introduce change of variables that reduces it to a form for which the standard results of the perturbation theory for linear operators [16] can be applied. Next, we derive perturbation formulae for the solutions of the latter problem. In Section 4, we use the derived perturbative formulae for the wavenumbers and mode functions to compute acoustic field by the formula (1), where the mode amplitudes $A_j$ are

computed using a wide-angle mode parabolic approximation for Equation (2). We show that even a single call of the spectral problem solver together with our perturbation theory allows to perform accurate simulation of sound propagation in a very large area.

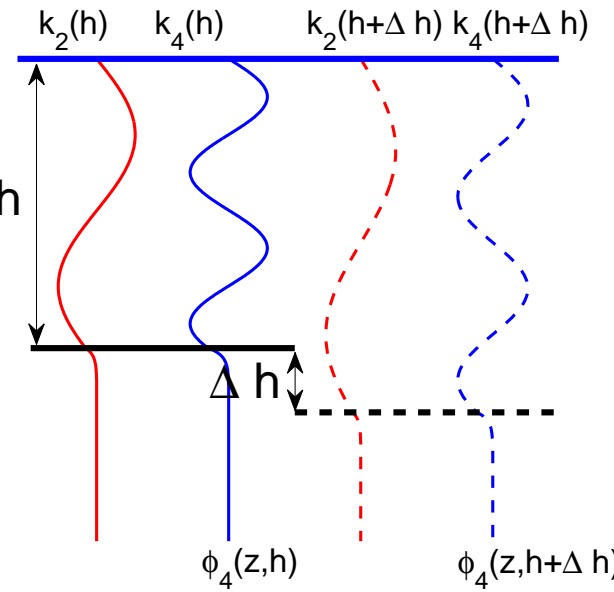

**Figure 1.** Perturbation of waterborne modes by water depth variation in a typical shallow-water waveguide with penetrable bottom.

## 2. Acoustical Spectral Problem

The main goal of the present study is to improve the computational efficiency of sound propagation models based on the normal modes theory that rely on representation of acoustic field in the form (1). In the course of the field computation, such models usually make numerous calls of a solver of the following acoustic spectral problem [1,2]:

$$
\begin{cases}
\dfrac{\mathrm{d}^2 \phi_j}{\mathrm{d}z^2} + \dfrac{\omega^2}{(c(z))^2} \phi_j = k_j^2 \phi_j, & z \in (0, h) \cup (h, H), \\[2mm]
\phi_j|_{z=0} = 0, \\[2mm]
\phi_j|_{z=H} = 0, \\[2mm]
\phi_j|_{z=h^-} = \phi_j|_{z=h^+}, \\[2mm]
\dfrac{1}{\rho} \dfrac{\mathrm{d}\phi_j}{\mathrm{d}z}\bigg|_{z=h^-} = \dfrac{1}{\rho} \dfrac{\mathrm{d}\phi_j}{\mathrm{d}z}\bigg|_{z=h^+},
\end{cases}
\tag{3}
$$

on an interval $z \in [0, H]$, where $F|_{z=h^\pm}$ denotes one-sided limits of some function $F(z)$ (possibly discontinuous) as $z$ approaches the point $z = h$ from below or above, respectively. The function $c(z)$ in (3) is assumed to be continuously differentiable on the intervals $I_1 = [0, h)$ and $I_2 = (h, H]$ (it can therefore have a finite jump discontinuity at $z = h$), and the density is a piecewise-constant function:

$$
\rho(z) = \begin{cases} \rho_w, & \text{for } z \leq h, \\ \rho_b, & \text{for } z > h. \end{cases}
\tag{4}
$$

The second and the third equalities in Equation (3) express pressure-release boundary conditions at the surface $z = 0$ and at some sub-bottom $z = H$ (the sub-bottom is introduced to avoid complications associated with continuous spectrum of the halfspace). The value of $H$ is chosen to be sufficiently large in order to ensure the convergence of the solution of the propagation problem of interest. The third and the fourth equalities are continuity

conditions for sound pressure and particle velocity at the interface $z = h$ between the water and the upper sediment layer of the bottom.

Solutions of the problem (3) are pairs $(k_j, \phi_j(z))$, where $k_j$ is called horizontal (modal) wavenumber, and $\phi_j(z)$ is the respective (vertical) mode function. The Dirichlet boundary conditions at the endpoints of the interval $[0, H]$ in Equation (3) ensure that the problem has purely discrete spectrum consisting of a countable set of real eigenvalues $k_j^2$, $j = 1, 2, \ldots$. We assume that they are ordered in such a way that $k_j^2 \geq k_{j+1}^2$ for all $j$. Note that the mode functions form an orthonormal basis with respect to the scalar product

$$(f, g) = \int_0^H \frac{f(z) g(z)}{\rho(z)} dz , \tag{5}$$

i.e., $(\phi_i, \phi_j) = \delta_{ij}$ (see [1,2]).

Hereafter, we are mainly interested in waterborne modes, i.e., such modes that their eigenfunctions $\phi_j(z)$ have maxima in the water column, and their eigenvalues $k_j$ belong to the interval $[\omega/c_b, \omega/c_{min}]$ ($c_b$ is the sound speed in the upper layer of the bottom, and $c_{min}$ is the minimum value of the sound speed in the water column). Away from the source the field is mostly formed by waterborne modes, and therefore in practical problems of underwater acoustics they are of primary importance.

In the literature, horizontal wavenumbers $k_j$ are often called eigenvalues of the problem (3) [1], although strictly speaking this term should be used only for their squares $k_j^2$. According to this tradition, hereafter we also call $k_j$ eigenvalues.

Note that the problem (3) can be considered a regular counterpart to the singular spectral problem for the Pekeris operator [17,18] (where the condition at $z = H$ is replaced by the boundedness condition at $z \to \infty$). Within this work, we restrict our attention to the self-adjoint case, i.e., to the case of lossless bottom (with negligible attenuation). The regular case considered below is arguably more important from the practical point of view.

From the standard theory of regular Sturm–Liouville problems (see, e.g., in [1,18,19]) it is known that eigenvalues of (3) are real of multiplicity 1, and that they form a countable set $k_1^2 > k_2^2 > \ldots$. The respective eigenfunctions $\phi_j(z)$ are obviously continuous on $[0, H]$, and their restrictions $\phi_j|_{I_{1,2}}$ to the intervals $I_{1,2}$ belong to spaces $C^2(I_{1,2})$, respectively.

In this case, $k_j$ and $\phi_j$ can be considered functions of $h$ (while the latter is in turn a function of horizontal coordinates $x, y$). An illustration of this dependence for a typical shallow-water waveguide with penetrable bottom is presented in Figure 1.

Observe that for the field computation by the formula (1) it is necessary to know $k_j(x, y)$ and $\phi_j(z, x, y)$ for all values of $x, y$ (or for all values of $h(x, y)$ along in the area of interest). This requires the solution of the problem (3) for all values of $h$ with certain sufficiently small step. On the other hand, we can consider the variations of water depth as a perturbation with respect to certain average value $h_0$. Expressing the bathymetry variations $h = h_0 + \Delta h(x, y)$ as fluctuations of water depth $\Delta h(x, y)$ around some average value $h_0$ we can use Taylor series expansion for horizontal wavenumbers and mode functions

$$k_j(x, y) \sim k_{j,0} + k'_{j,0} \Delta h(x, y) + \frac{1}{2} k''_{j,0} (\Delta h(x, y))^2 + \ldots , \tag{6}$$

$$\phi_j(z, x, y) \sim \phi_{j,0}(z) + \phi'_{j,0}(z) \Delta h(x, y) + \frac{1}{2} \phi''_{j,0}(z) (\Delta h(x, y))^2 + \ldots , \tag{7}$$

where prime denotes the derivative of $k_j$ or $\phi_j(z)$ (for a fixed value of $z = z_r$) with respect to the parameter $h$.

## 3. Perturbation of Normal Modes by the Bathymetry Variations

The problem for the bathymetry perturbation (i.e., interface perturbation) can be reduced to a well-studied problem of the potential perturbation in the stationary Schrödinger

equation [11]. Let us introduce new variable $t = zh_0/h$ and denote $T = Hh_0/h$. After substitution into Equation (3), we obtain the following spectral problem:

$$
\begin{cases}
\dfrac{\mathrm{d}^2 \psi_j}{\mathrm{d}t^2} + \dfrac{h^2 \omega^2}{h_0^2 (c(t))^2} \psi_j = \dfrac{h^2}{h_0^2} k_j^2 \psi_j\,, \\[2mm]
\psi_j|_{t=0} = 0\,, \\[2mm]
\psi_j|_{t=T} = 0\,, \\[2mm]
\psi_j|_{t=h_0^-} = \psi_j|_{t=h_0^+}\,, \\[2mm]
\dfrac{1}{\rho} \dfrac{\mathrm{d}\psi_j}{\mathrm{d}t}\bigg|_{t=h_0^-} = \dfrac{1}{\rho} \dfrac{\mathrm{d}\psi_j}{\mathrm{d}t}\bigg|_{t=h_0^+}\,,
\end{cases} \tag{8}
$$

where $k_j$ are the same wavenumbers as those obtained by solving (3), and eigenfunctions $\psi_j(t)$ are related to the eigenfunctions of the original problem (3) by the relation $\phi_j(z) = \sqrt{\dfrac{h_0}{h}} \psi_j(t)$. Denoting $\epsilon = \Delta h/h_0 = (h - h_0)/h_0$ for convenience, we introduce the following expansions:

$$
h = h_0(1 + \epsilon)\,,
$$

$$
T = \frac{Hh_0}{h} = H\left(1 - \epsilon + \frac{\epsilon^2}{2}\right)\,,
$$

$$
k_j = k_j^{(0)} + \epsilon k_j^{(1)} + \frac{\epsilon^2}{2!} k_j^{(2)} + \dots\,,
$$

$$
\psi_j = \psi_j^{(0)} + \epsilon \psi_j^{(1)} + \frac{\epsilon^2}{2!} \psi_j^{(2)} + \dots \tag{9}
$$

$$
Q(z) = Q(t) + \epsilon t Q'(t) + \epsilon^2 \frac{t^2 Q''(t)}{2!} + \dots\,. \tag{10}
$$

where $Q = \dfrac{\omega^2}{c^2}$, and superscripts in parentheses stand for derivatives with respect to $\epsilon$.

Now, we follow the standard scheme of the perturbation theory. Substituting expansions (9) into (8) and separating terms with identical powers of epsilon $\epsilon$, we obtain for $\epsilon^0$ the spectral problem (8) for $k_j^{(0)}$ and $\psi_j^{(0)}(t)$ with the boundary condition $\psi_j^{(0)}|_{t=H} = 0$.

For $\epsilon^1$ we obtain

$$
\frac{\mathrm{d}^2 \psi_j^{(1)}}{\mathrm{d}t^2} + Q\psi_j^{(1)} + (tQ' + 2Q)\psi_j^{(0)} = \left(k_j^{(0)}\right)^2 \psi_j^{(1)} + 2\left(\left(k_j^{(0)}\right)^2 + 2k_j^{(0)}k_j^{(1)}\right)\psi_j^{(0)}\,. \tag{11}
$$

In order to obtain $k_j^{(1)}$ and $\psi_j^{(1)}$ from the latter equation, we multiply it by $\psi_j^{(0)}$ and by $\psi_i^{(0)}$ (in the sense of the scalar product). As these functions are orthogonal on $[0, H]$, we have to transfer the boundary condition from $t = T$ to $t = H$.

$$
\psi_j^{(0)} + \epsilon \psi_j^{(1)} + \frac{\epsilon^2}{2}\psi_j^{(2)} + \frac{\epsilon^3}{6}\psi_j^{(3)}|_{t=T} = \psi_j^{(0)}|_{t=H} + \epsilon\left(\psi_j^{(1)} - H\frac{\mathrm{d}\psi_j^{(0)}}{\mathrm{d}t}\right)\Bigg|_{t=H} +
$$

$$
\epsilon^2\left(\frac{\psi_j^{(2)}}{2} - H\frac{\mathrm{d}\psi_j^{(0)}}{\mathrm{d}t} + \dots\right)\Bigg|_{t=H} + \epsilon^3\left(\frac{\psi_j^{(3)}}{6} - H\frac{\mathrm{d}\psi_j^{(0)}}{\mathrm{d}t} - \frac{H^3}{6}\frac{\mathrm{d}^3\psi_j^{(0)}}{\mathrm{d}t^3} + \frac{H^2}{2}\frac{\mathrm{d}^2\psi_j^{(1)}}{\mathrm{d}t^2} + \dots\right)\Bigg|_{t=H} = 0\,, \tag{12}
$$

where dots denote the terms containing $\psi_j^{(0)}, \psi_j^{(1)}, \psi_j^{(2)}$ and their derivatives that vanish according to the following identities:

$$
\left.\frac{\mathrm{d}^2\psi_j^{(0)}}{\mathrm{d}t^2}\right|_{t=H} = \left(\left(k_j^{(0)}\right)^2 - Q\right)\psi_j^{(0)}\Big|_{t=H} = 0, \quad \left.\frac{\mathrm{d}\psi_j^{(1)}}{\mathrm{d}t}\right|_{t=H} = H\left.\frac{\mathrm{d}^2\psi_j^{(0)}}{\mathrm{d}t^2}\right|_{t=H} = 0, \text{ etc.} \tag{13}
$$

Separating in (12) terms of different powers of $\epsilon$, we obtain a family of boundary conditions for $\psi_j^{(0)}, \psi_j^{(1)}, \psi_j^{(2)}, \psi_j^{(3)}$ that read as

$$
\psi_j^{(0)}\Big|_{t=H} = 0, \quad \psi_j^{(1)}\Big|_{t=H} = H\left.\frac{\mathrm{d}\psi_j^{(0)}}{\mathrm{d}t}\right|_{t=H}, \quad \psi_j^{(2)}\Big|_{t=H} = -H\left.\frac{\mathrm{d}\psi_j^{(0)}}{\mathrm{d}t}\right|_{t=H},
$$

$$
\psi_j^{(3)}\Big|_{t=H} = \left(6H - 2H^3\left(\left(k_j^{(0)}\right)^2 - Q\right)\right)\left.\frac{\mathrm{d}\psi_j^{(0)}}{\mathrm{d}t}\right|_{t=H}. \tag{14}
$$

Now, let us compute the scalar product of Equation (11) with the function $\psi_j^{(0)}$

$$
\int_0^H \left(\frac{\mathrm{d}^2\psi_j^{(1)}}{\mathrm{d}t^2} + Q\psi_j^{(1)} + (tQ' + 2Q)\psi_j^{(0)}\right)\frac{\psi_j^{(0)}}{\rho(t)}\mathrm{d}t =
$$

$$
\left(k_j^{(0)}\right)^2 \int_0^H \frac{\psi_j^{(1)}\psi_j^{(0)}}{\rho(t)}\mathrm{d}t + 2\left(\left(k_j^{(0)}\right)^2 + k_j^{(0)}k_j^{(1)}\right)\int_0^H \frac{\psi_j^{(0)}\psi_j^{(0)}}{\rho(t)}\mathrm{d}t, \tag{15}
$$

using the following standard orthogonality identities (see, e.g., in [11])

$$
\int_0^H \frac{\psi_i^{(0)}\psi_j^{(0)}}{\rho(t)}\mathrm{d}t = \delta_{ij}, \quad \int_0^H \frac{\psi_j^{(k)}\psi_j^{(0)}}{\rho(t)}\mathrm{d}t = 0 \text{ for } k > 0. \tag{16}
$$

Integrating the first term on the right-hand side of Equation (15) by parts and using (16) and (14), we obtain the following expression for $k_j^{(1)}$

$$
k_j^{(1)} = -k_j^{(0)} + \frac{1}{k_j^{(0)}}\int_0^H Q(t)\frac{\left(\psi_j^{(0)}(t)\right)^2}{\rho(t)}\mathrm{d}t + \frac{1}{2k_j^{(0)}}\int_0^H tQ'(t)\frac{\left(\psi_j^{(0)}(t)\right)^2}{\rho(t)}\mathrm{d}t + \frac{\mathcal{B}_{jj}}{2k_j^{(0)}}, \tag{17}
$$

where we introduced the following notation:

$$
\mathcal{B}_{ji} = -\left(\frac{H}{\rho(t)}\frac{\mathrm{d}\psi_j^{(0)}}{\mathrm{d}t}\frac{\mathrm{d}\psi_i^{(0)}}{\mathrm{d}t}\right)\Bigg|_{t=H} \tag{18}
$$

for the terms associated with the sub-bottom stretching in new coordinate $t$ (obviously, these terms can be neglected for trapped/waterborne modes).

Multiplying Equation (11) by the function $\psi_i^{(0)}$ and repeating the steps described above, we arrive at the following formula for the scalar product of $\psi_j^{(1)}$ and $\psi_i^{(0)}$:

$$
\left(\left(k_j^{(0)}\right)^2 - \left(k_i^{(0)}\right)^2\right)\int_0^H \frac{\psi_j^{(1)}\psi_i^{(0)}}{\rho(t)}\mathrm{d}t = 2\int_0^H Q\frac{\psi_i^{(0)}\psi_j^{(0)}}{\rho(t)}\mathrm{d}t + \int_0^H tQ'\frac{\psi_i^{(0)}\psi_j^{(0)}}{\rho(t)}\mathrm{d}t + \mathcal{B}_{ji}. \tag{19}
$$

As $\{\psi_i^{(0)}\}$ form an orthogonal basis, the function $\psi_j^{(1)}$ can be expressed in the form of the following series:

$$\psi_j^{(1)} = \sum_{i \neq j} \frac{V_{ji}}{\left(k_j^{(0)}\right)^2 - \left(k_i^{(0)}\right)^2} \psi_i^{(0)} , \tag{20}$$

$$V_{ji} = 2 \int_0^H Q \frac{\psi_i^{(0)} \psi_j^{(0)}}{\rho(t)} \, dt + \int_0^H tQ' \frac{\psi_i^{(0)} \psi_j^{(0)}}{\rho(t)} \, dt + \mathcal{B}_{ji} . \tag{21}$$

Functions $\psi^{(1)}$ by their construction are the derivatives $\frac{\partial \psi}{\partial \epsilon} = h_0 \frac{\partial \psi}{\partial h}$. Consequently, the derivatives $\frac{\partial \phi}{\partial h}$ of eigenfunctions $\phi(z)$ of the original problem (3) can be found as

$$\left. \frac{\partial \phi_j}{\partial h} \right|_{h=h_0} = \frac{1}{h_0} \left( -\frac{\psi_j^{(0)}}{2} - t \frac{\partial \psi_j^{(0)}}{\partial t} + \psi_j^{(1)} \right) \Bigg|_{t=z} . \tag{22}$$

Let us now proceed with the terms of the order $\epsilon^2$ that arise when substituting (9) into (8). We obtain the following equation:

$$\frac{d^2 \psi_j^{(2)}}{dt^2} + Q\psi_j^{(2)} + 2(tQ' + 2Q)\psi_j^{(1)} + (t^2 Q'' + 4tQ' + 2Q)\psi_j^{(0)} =$$
$$\left(k_j^{(0)}\right)^2 \psi_j^{(2)} + 4\left(\left(k_j^{(0)}\right)^2 + k_j^{(0)} k_j^{(1)}\right)\psi_j^{(1)} + 2\left(\left(k_j^{(0)}\right)^2 + 4k_j^{(0)} k_j^{(1)} + k_j^{(0)} k_j^{(2)} + \left(k_j^{(1)}\right)^2\right)\psi_j^{(0)} . \tag{23}$$

Multiplying it by $\psi_j^{(0)}$ (in the sense of the scalar product), and using the orthogonality relations (16) and boundary conditions (12), we obtain an expression for $k_j^{(2)}$,

$$k_j^{(2)} = -k_j^{(0)} - \frac{\left(k_j^{(1)}\right)^2}{k_j^{(0)}} - 4k_j^{(1)} + \frac{1}{k_j^{(0)}} \int_0^H (tQ' + 2Q) \frac{\psi_j^{(1)} \psi_j^{(0)}}{\rho} \, dt$$

$$+ \frac{1}{k_j^{(0)}} \int_0^H \left(t^2 \frac{Q''}{2} + 2tQ' + Q\right) \frac{\left(\psi_j^{(0)}\right)^2}{\rho} \, dt - \frac{\mathcal{B}_{jj}}{k_j^{(0)}} . \tag{24}$$

The scalar product of the equality (23) and $\psi_i^{(0)}$ leads to the expression for $\psi_j^{(2)}$

$$\psi_j^{(2)} = \sum_{i \neq j} \frac{W_{ji}}{\left(k_j^{(0)}\right)^2 - \left(k_i^{(0)}\right)^2} \psi_i^{(0)} , \quad \text{where}$$

$$W_{ji} = -4k_j^{(0)} \left(k_j^{(0)} + k_j^{(1)}\right) \frac{V_{ji}}{\left(k_j^{(0)}\right)^2 - \left(k_i^{(0)}\right)^2}$$

$$+ \int_0^H \left(2Q + 4tQ' + t^2 Q''\right) \frac{\psi_j^{(0)} \psi_i^{(0)}}{\rho} \, dt + 2 \int_0^H \left(2Q + tQ'\right) \frac{\psi_j^{(1)} \psi_i^{(0)}}{\rho} \, dt - 2\mathcal{B}_{ji} , \tag{25}$$

while the second derivative of the eigenfunction of the original problem (3) can be found as

$$\left. \frac{\partial^2 \phi_j}{\partial h^2} \right|_{h=h_0} = \frac{1}{h_0^2} \left( \frac{3\psi_j^{(0)}}{4} + 3t \frac{\partial \psi_j^{(0)}}{\partial t} - \psi_j^{(1)} + t^2 \frac{\partial^2 \psi_j^{(0)}}{\partial t^2} - 2t \frac{\partial \psi_j^{(1)}}{\partial t} + \psi_j^{(2)} \right) \Bigg|_{t=z} . \tag{26}$$

Let us conclude this section with the formula for $k_j^{(3)}$, that can be obtained in by reiterating the calculations above:

$$k_j^{(3)} = -6\left(k_j^{(1)} + k_j^{(2)} + \frac{\left(k_j^{(1)}\right)^2}{k_j^{(0)}} + \frac{k_j^{(1)}k_j^{(2)}}{2k_j^{(0)}}\right) + \frac{1}{k_j^{(0)}}\int_0^H \left(\frac{3t}{2}Q' + 3Q\right)\frac{\psi_j^{(2)}\psi_j^{(0)}}{\rho}\,dt$$

$$+ \frac{1}{k_j^{(0)}}\int_0^H \left(\frac{3t^2}{2}Q'' + 6tQ' + 3Q\right)\frac{\psi_j^{(1)}\psi_j^{(0)}}{\rho}\,dt$$

$$+ \frac{1}{k_j^{(0)}}\int_0^H \left(\frac{t^3}{2}Q''' + 3t^2Q'' + 3tQ'\right)\frac{\left(\psi_j^{(0)}\right)^2}{\rho}\,dt + \left(3 - H^2\left(\left(k_j^{(0)}\right)^2 - Q\right)\right)\frac{\mathcal{B}_{jj}}{k_j^{(0)}}. \quad (27)$$

### 4. Numerical Example: the 3D Coastal Wedge Problem

Consider a coastal wedge shown in Figure 2 with the opening angle $\alpha = 2.86°$ and water depth near the source $h = 200$ m. Assume that the sound speed and the density in the water column are $c_1 = 1500$ m/s and $\rho_1 = 1$ g/cm³, respectively, and the values of these parameters for the bottom are $c_2 = 1700$ m/s and $\rho_2 = 1.5$ g/cm³. Sound propagation in the wedge with the parameters specified above is considered a standard benchmark problem for 3D models of sound propagation in underwater acoustics (see, e.g., in [1,20]).

A Cartesian coordinate system in the waveguide is chosen in such a way that $x$ axis is aligned along the isobath, while the direction of the $y$ axis coincides with the water depth gradient, and a point source of the frequency $f = 25$ Hz, is deployed at the depth $z_s = 100$ m (its horizontal coordinates are $x_s = 0$, $y_s = 0$).

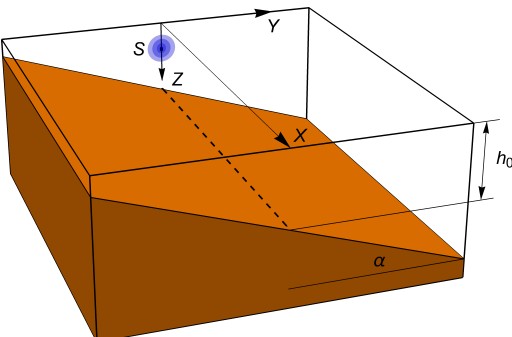

**Figure 2.** Coastal wedge with penetrable bottom.

The problem of sound propagation in the coastal wedge with the parameters specified above is usually used for the validation of various mathematical models in shallow-water acoustics. It was recently shown that an accurate solution of this problem in the cross-slope direction can be obtained under the adiabaticity assumption [21]. In our opinion, this result is remarkable, as it indicates that many propagation problems can be successfully tackled by a computationally lightweight adiabatic models.

The main idea of the present study is that the efficiency of the respective approaches can be further improved if we reduce the number of calls of the Sturm–Liouville problems solver. The first thing to do in this direction is to answer the question on how far can we go from the source with just one call of the normal modes computation. Arguably, the wedge problem can be considered a worst case scenario for the application of our perturbation theory, as depth variations in this environment are by no means small. Indeed, an accurate solution of the wedge problem reported in [21] required the computation of modes for the values of water depth in the range 50–350 m, i.e., and therefore we have $h_0 = 200$ m, and $\Delta h = -150 \cdots + 150$ m.

Figure 3 shows the dependencies of the wavenumbers $k_j$ on the water depth $h$ for all three waterborne modes excited by the source. The red lines correspond to the "exact" eigenvalue dependencies on $h$. Approximations of these dependencies obtained by the perturbation formulae of the first, second, and third order are represented by magenta,

blue, and black lines, respectively. It can be seen that the 3rd-order approximation is almost perfect for $\Delta h \in [-100 \text{ m}, 100 \text{ m}]$ for the first and second modes, and also works pretty well for $\Delta h \in [-50 \text{ m}, 100 \text{ m}]$ for the third mode. The 2nd-order perturbation theory provides accurate approximations of the wavenumber variations for $|\Delta h| \leq 50$ m, while the linear (1st-order) approximation of $k_j(h)$ ensures reasonable quality only for $|\Delta h| \leq 10$–20 m. Note that all approximations are clearly invalid beyond the cut-off depth value. Yet, high-order approximations offer considerable improvement over the accuracy of the linear one. It is also important that the calculations of the derivatives $k_j^{(n)}$ by the perturbative formulae require almost no extra computational cost (in addition to the solution of the "unperturbed" Sturm–Liouville problem).

Although the accuracy of approximation of the eigenvalues $k_j$ dependence on $h$ can be of interest per se, it is more important to investigate the accuracy of the field computation if our formulae are used to compute $\phi_j(z, x, y)$ in the expansion (1) and $k_j(x, y)$ in Equation (2). In this study, we compute the solution of the wedge problem using the perturbation theory for the normal modes in combination with the field representation (1) and the pseudodifferential parabolic equations for mode amplitudes $A_j$ (PDMPEs). The latter equations can be used to compute the solution of Equation (2) neglecting the back-scattering (which is known to be small in the case of the penetrable wedge), see in [21,22] for the details.

The solution of the wedge problem obtained by the scheme outlined here is obtained under several assumptions and approximation. In order to validate its accuracy, we compare it against the solution computed using source images technique [20]. The comparison along the $x$ axis for $y = 0$ and $z = z_r = 30$ m is presented in Figure 4. From Figure 4d, it is clear that the solution based on the PDMPE theory and adiabatic approximation (we actually use a wide-angle parabolic approximation for Equation (2) and the expansion (1) according to the theory from in [21,22]). exhibits excellent agreement with the reference solution if the spectral problem is solver for each value of $h$ (for all $x$). If we solve it only for $h = h_0$ and use first-order perturbative formulae, then the field can be computed accurately only for $x < 6$–7 km (see Figure 4a). A significant improvement can be achieved by employing 2nd-order formula (Figure 4b), and the agreement with the reference solution in this case is very good up to $x = 15$ km, which is already sufficient for many practical problems arising in underwater acoustics, where the speed of the field simulation is of primary importance. Note that in this case the 3rd-order formula extends the correct solution by a couple of kilometers in range, as shown in Figure 4c. As for $h < 100$ m (i.e., for $|\Delta h| > 100$ m), even the third-order approximation for the wavenumbers $k_j(h)$ fails to reproduce actual dependence (see Figure 3), the solution for $x > 17$ km cannot be accurate. Indeed, it is known that interference pattern along this interval is formed by the field of the first mode that reaches this area by "direct" and "refracted-by-the-wedge" paths [23,24], and the turning point of the latter is located very close to the wedge apex, where perturbation theory does not provide sufficient accuracy for the wavenumbers.

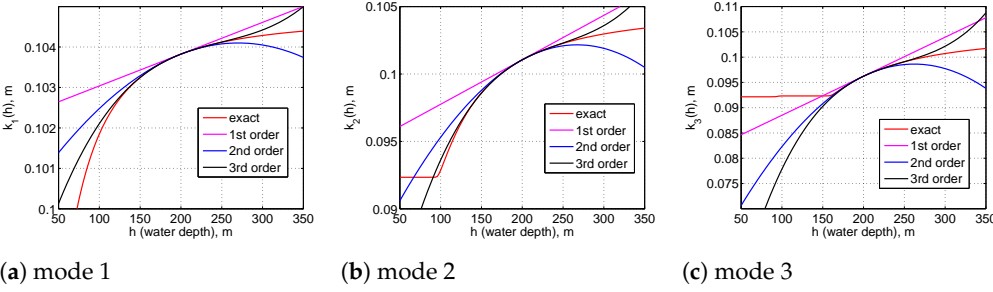

(**a**) mode 1        (**b**) mode 2        (**c**) mode 3

**Figure 3.** Dependence of $k_j(h)$ in the coastal wedge and its approximations by perturbative formulae.

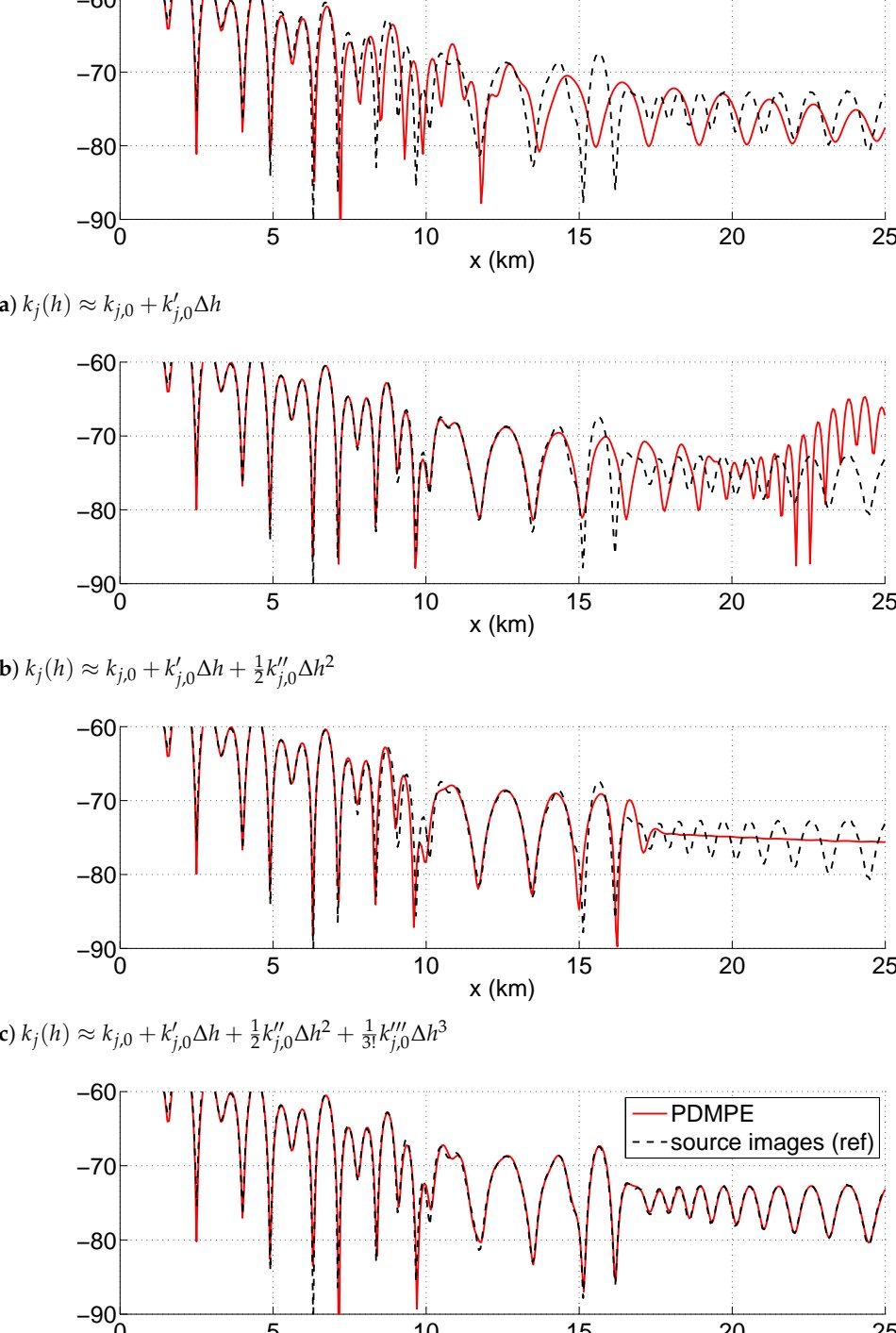

**(a)** $k_j(h) \approx k_{j,0} + k'_{j,0}\Delta h$

**(b)** $k_j(h) \approx k_{j,0} + k'_{j,0}\Delta h + \frac{1}{2}k''_{j,0}\Delta h^2$

**(c)** $k_j(h) \approx k_{j,0} + k'_{j,0}\Delta h + \frac{1}{2}k''_{j,0}\Delta h^2 + \frac{1}{3!}k'''_{j,0}\Delta h^3$

**(d)** Exact values of $k_j$ for each $h$

**Figure 4.** Acoustical field magnitude (in dB re 1 m from the source) in the coastal wedge as a function of $x$ for $y = 0$, $z = 30$ m computed using pseudodifferential mode parabolic equations and (1) (red solid line). The reference solution by the method of source images is shown by the black dashed line.

The slice of acoustical field $P(x, y, z)$ by the horizontal plane $z = z_r = 30$ m for the normal modes computed by the 3rd-order perturbation theory and the respective reference solution are shown in Figure 5.

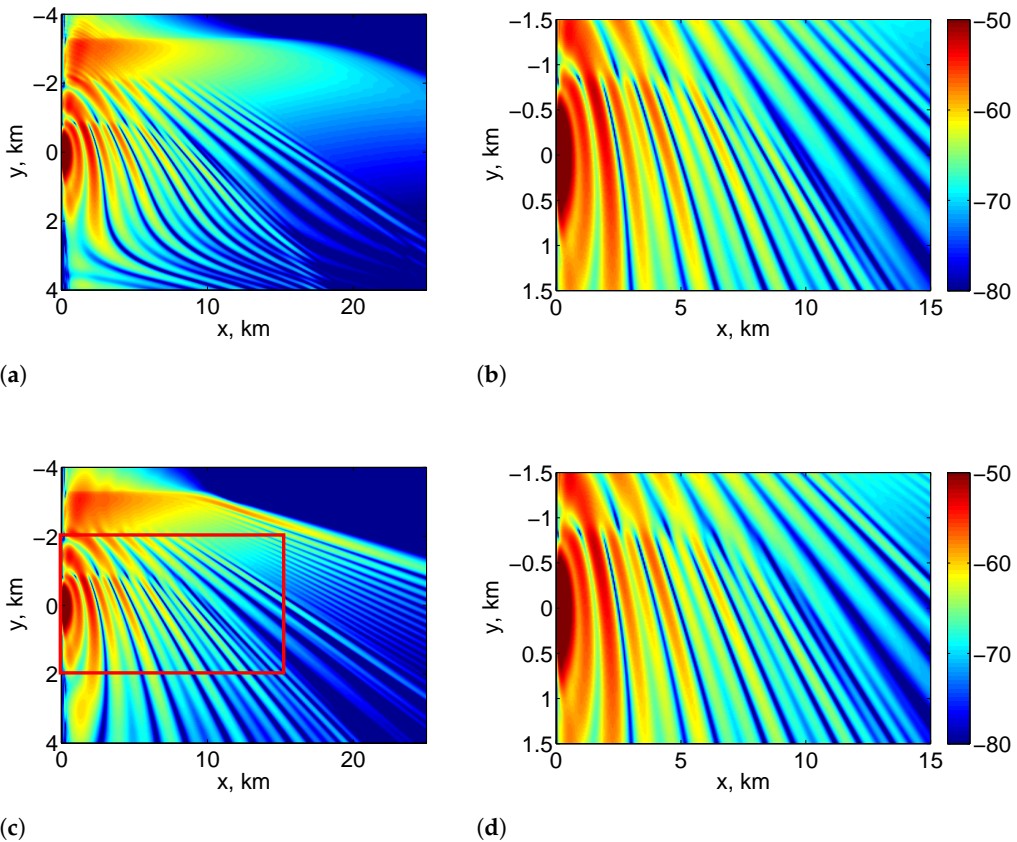

**Figure 5.** The field at $z = z_r$ computed using eigenvalues and eigenfunctions obtained by the 3rd-order perturbation formulae (**a**,**b**) and by solving the Sturm–Liouville problem (3) for each $h$ (**c**,**d**).

It can be seen that within the rectangle $|y| \leq 2\,\text{km}, x \leq 15\,\text{km}$ that the solution computed using $k_j$ and $\phi_j$ approximated by the formulae (6), (7) cannot be distinguished from the reference solution. At the same time, the perturbation theory allows to reduce the computational cost by a factor of 10 in this case.

The simulation results confirm that the perturbation theory for normal modes developed in our study can be successfully used to improve the efficiency of the sound propagation model based representation of acoustical field in the form (1), and that sometimes it is sufficient to solve the spectral problem (3) only once for some average water depth value $h_0$.

We also observed that even if the accuracy provided by the perturbation theory and a single call of mode solver is not sufficient, one can use solve the spectral problem for maximal and minimal values of the depth, compute the derivatives of wavenumbers and mode functions by our formulae, and then use a clamped spline approximation for the entire area of interest. For example, in the wedge case this results in a solution that perfectly coincides with the one obtained by using $k_j$ computed for all values of the water depth. This two-point clamped spline approach can be considered a good compromise between the efficiency and the accuracy.

## 5. Conclusions

In this study, formulae for the first- and second-order derivatives of wavenumbers and eigenfunctions of normal modes in a shallow-water waveguide with respect to the water depth are obtained (we stress again that main formulae were outlined in [15], but the detailed derivation was not presented there). They are derived by using vertical coordinate transformation that leads to the reformulation of the problem in such a way that the

interface perturbation turns into the potential perturbation in a stationary Schrödinger equation. Clearly, our results can be generalized to the interface perturbation theory of arbitrary order. On the practical side, however, most problems can be covered by the second- and third-order formulae presented here.

Note that first-order derivatives of wavenumbers and mode functions with respect to $h$ were computed by a different method in [9,12,13], however unfortunately the latter approach cannot be generalized to obtain higher-order formulae. It can be shown that our expression for the first derivative (20) can be reduced to the respective formula from [9]. However, the two expressions are absolutely different from the computational point of view. Indeed, in our case the main contribution to the first derivative of the eigenfunction with respect to the $h$ parameter is made by the derivatives of the latter with respect to $z$, and the series over other unperturbed eigenfunctions can be considered a small correction to the latter. By contrast, the expression for the eigenfunction perturbation derived by Trofimov is a pure expansion over unperturbed eigenfunctions. As such, it is much less robust, as the convergence of the series is quite slow.

In this study, we presented an example where the acoustical field is computed in a 3D wedge benchmark problem. Even for this idealized scenario the computational time can be reduced by a factor of 10 using our perturbative formulae. In more realistic problems, the reduction in the computational cost can be even more significant, especially when it is necessary to take mode coupling effects into account.

**Author Contributions:** Methodology, P.P. and M.T.; formal analysis, A.Z. and P.P.; investigation, A.Z. and P.P. All authors have read and agreed to the published version of the manuscript.

**Funding:** This study was supported by POI FEB RAS Program "Modeling of various-scale dynamical processes in the ocean" (project No. 121021700341-2).

**Institutional Review Board Statement:** Not applicable.

**Informed Consent Statement:** Not applicable.

**Conflicts of Interest:** The authors declare no conflict of interest.

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
