# Peer review of "Improving the Performance of Mode-Based Sound Propagation Models by Using Perturbation Formulae for Eigenvalues and Eigenfunctions"

_jmse, doi:10.3390/jmse9090934_

Round 1
Reviewer 1 Report
Review of manuscript JMSE-1314254
The manuscript describes expressions for normal modes that allow a more efficient numerical analysis by reducing the number of normal modes to be considered in acoustic propagation in shallow water, based on the analysis of the variation of eigenvalues and eigenfunctions for different depths.
The manuscript is well structured and well written. The results appear to be consistent with the objectives and show a significant reduction in the number of modes needed to compute the acoustic field in a waveguide of varying height.
Here are some comments that may help to improve the quality of the manuscript.
1) It is a matter of style, but perhaps the use of "we" in the text could be avoided.
2) Even if an experienced reader is familiar with the geometrical description and the acoustic propagation problem discussed in the manuscript, and can be found in cites [1-3], it would not hurt to spend a few lines explaining what the authors consider the "3D ocean waveguide" and the wave equation to which the eigenvalues and eigenvectors refer. Likewise, the coordinate system used. Only in the Abstract and in lines 22-23 it is mentioned that (x,y) are the horizontal axes. Finally, the definition of axes is in lines 111-113 and Fig 2.
3) Also, I don’t find the need of paragraph in lines 27bis to 29, with Eqs. (3) and (4): another example of what? In the mentioned equations, the function ρ(z), which is found in the denominator, has not been introduced until page 4.
4) There is an extra right parenthesis in line 78bis.
5) Waterborne modes, mentioned in the caption of Fig. 1 and in the paragraph in lines 80-81, need a formal definition, rather than a description in parentheses. Why those are the modes of greatest interest also deserves a text line.
6) Eq. (11): Superscripts in parentheses stand for derivatives with respect to the new variable epsilon? This is not pointed until Eq. (22). Even if it could be guessed from the series expansion, the explicit mention would help to follow the development.
7) Line 102: “propblem” instead of problem.
8) The example chosen in section 4 may seem arbitrary. A brief explanation of the choice of values would not be superfluous. If the reason is to compare with previous results already published, it seems mandatory to cite them. To understand the location of the point source, the reader has to read the paragraphs from lines 107 to 111. Please consider adding a sentence with the coordinates of the source position. Finally, a mention of the application of the method to other possible values would also be interesting.
9) The first three sentences of the paragraph starting in line 120 (lines 120-123) could be at the beginning of Section 2 or, at least, earlier in the manuscript. Because Sections 2 and 3 contain complex mathematical developments in the spectral domain and it is difficult to the reader to grasp the purpose of the analysis and how the obtained equations can be used or applied to the involved problem.
10) In Fig. 5, a color dB scale is missing.
Author Response
Dear Reviewer! Thank You very much for Your kind attention to our manuscript and Your comments. Below we provide a point-by-point response to Your comments/suggestions.
1)
Comment: It is a matter of style, but perhaps the use of "we" in the text could be avoided.
Response: Although it is no secret that many people prefer impersonal sentences, in this case we prefer to keep "we". It seems that sentences of this form do not violate any rules (see, e.g., https://oxfordediting.com/to-we-or-not-to-we-the-first-person-in-academic-writing/). One of the authors is also a big fan of Umberto Eco and mostly follows his "How to Write a Thesis" manual in his scientific writing. Hopefully our reviewer would understand our decision, and we are grateful for this suggestion anyway.
2)
Comment: Even if an experienced reader is familiar with the geometrical description and the acoustic propagation problem discussed in the manuscript, and can be found in cites [1-3], it would not hurt to spend a few lines explaining what the authors consider the "3D ocean waveguide" and the wave equation to which the eigenvalues and eigenvectors refer. Likewise, the coordinate system used. Only in the Abstract and in lines 22-23 it is mentioned that (x,y) are the horizontal axes. Finally, the definition of axes is in lines 111-113 and Fig 2.
Response: A brief explanation of what is an oceanic waveguide is added and a reference is provided. The coordinates are also introduced just before Eq. (1).
3)
Comment: Also, I don’t find the need of paragraph in lines 27bis to 29, with Eqs. (3) and (4): another example of what? In the mentioned equations, the function ρ(z), which is found in the denominator, has not been introduced until page 4.
Response: The specified paragraph was removed according to the reviewer's suggestion.
4)
Comment: There is an extra right parenthesis in line 78bis.
Response: Extra parenthesis is removed.
5)
Comment: Waterborne modes, mentioned in the caption of Fig. 1 and in the paragraph in lines 80-81, need a formal definition, rather than a description in parentheses. Why those are the modes of greatest interest also deserves a text line.
Response: The formal definition is provided, and a brief explanation of the importance of waterborne modes is added.
6)
Comment: Eq. (11): Superscripts in parentheses stand for derivatives with respect to the new variable epsilon? This is not pointed until Eq. (22). Even if it could be guessed from the series expansion, the explicit mention would help to follow the development.
Response: The reviewer is right, the superscripts stand for the derivatives with respect to \epsilon. The explanation is added just after the point where \epsilon is introduced.
7)
Comment: Line 102: “propblem” instead of problem.
Response: The issue is fixed.
8)
Comment: The example chosen in section 4 may seem arbitrary. A brief explanation of the choice of values would not be superfluous. If the reason is to compare with previous results already published, it seems mandatory to cite them. To understand the location of the point source, the reader has to read the paragraphs from lines 107 to 111. Please consider adding a sentence with the coordinates of the source position. Finally, a mention of the application of the method to other possible values would also be interesting.
Response: Source coordinates are specified explicitly. The wedge with the parameters used in our study is a standard benchmark problem for 3D models of sound propagation. An explanation and citations are added in the first paragraph of Sec. 4.
9)
Comment: The first three sentences of the paragraph starting in line 120 (lines 120-123) could be at the beginning of Section 2 or, at least, earlier in the manuscript. Because Sections 2 and 3 contain complex mathematical developments in the spectral domain and it is difficult to the reader to grasp the purpose of the analysis and how the obtained equations can be used or applied to the involved problem.
Response: The beginning of Sec. 2 is expanded according to the reviewer's suggestion.
10)
Comment: In Fig. 5, a color dB scale is missing.
Response: Colorbar was added in Fig. 5.
Reviewer 2 Report
Numerical simulation of underwater sound propagation is a key tool for the prediction of anthropogenic impacts and underwater acoustical detection of objects or communications. Studies addressed to reduce the computational cost of well stablished calculation methods are necessary and welcome.
The present work presents a perturbative approximation common in other fields of pyhsics, contributing to improve the computational efficency of the normal mode method and evaluating the obtained accuracy, permitting to balance between the reduction of computing time and the validity of results. It is remarkebly well written and it presents the results in an adequate form. I suggest the publication in its present form with only two typing corrections:
-x distance units in figure 4 would better be inside parenthesis: x(km).
- Also in fig 4 (b), an h seems to lack in the second term of (b)
k j ( h ) ≈ k j,0 + k 0 j,0 ∆h + 2 1 k 00 j,0 ∆h 2
Author Response
Dear Reviewer! Thank You very much for Your kind attention to our manuscript and Your comments. Below we provide a point-by-point response to Your comments/suggestions.
1)
Comment: x distance units in figure 4 would better be inside parenthesis: x(km)
Response: Corrected according to the Reviewer's suggestion.
2)
Comment: Also in fig 4 (b), an h seems to lack in the second term of (b)
Response: Caption of Fig. 4b was fixed, missing h is added.
Reviewer 3 Report
This work studies a mode-propagation model in the 3D ocean environment using the perturbation theory for the local modes representation. The introduction is sufficiently provided and the conclusion is well described based on the appropriate numerical results. Some minor comments have been given as below. If these issues will be resolved, there will be no critical problems in the publication of this manuscript.
1) In Eq(12), the 'small O' is used. Please check it again.
2) Fix the caption of Fig. 4(b). 'h' is missing.
3) Why don't the authors provide the result for 3rd-order formula in Fig. 4? It is recommended to add it. And, please add some physical reasons on the disagreement of perturbation-based model of the authors above 15 km.
4) (Line 151-153) What is the solution with PDMPE theory and adiabatic approximation? Do the authors solve Eq. (2) and (5) based Eq. (1). Specify it.
Author Response
Dear Reviewer! Thank You very much for Your kind attention to our manuscript and Your comments. Below we provide a point-by-point response to Your comments/suggestions.
1)
Comment: In Eq(12), the 'small O' is used. Please check it again
Response: Eq. (12) is corrected.
2)
Comment: Fix the caption of Fig. 4(b). 'h' is missing.
Response: Caption of Fig. 4b was fixed.
3)
Comment: Why don't the authors provide the result for 3rd-order formula in Fig. 4? It is recommended to add it. And, please add some physical reasons on the disagreement of perturbation-based model of the authors above 15 km.
Response: A subplot with 3rd-order formula is added to Fig. 4. Physical reasons for the disagreement above 15 km are explained briefly in Sec. 4 (text highlighted in red).
4)
Comment: (Line 151-153) What is the solution with PDMPE theory and adiabatic approximation? Do the authors solve Eq. (2) and (5) based Eq. (1). Specify it.
Response: An explanation is provided and suitable references are given.